# POINTSAGE : MESH-INDEPENDENT SUPERRESOLUTION APPROACH TO FLUID FLOW PREDICTIONS

## ABSTRACT

Computational Fluid Dynamics (CFD) serves as a powerful tool for simulating fluid flow across diverse industries. High-resolution CFD simulations offer valuable insights into fluid behavior and flow patterns. As resolution increases, computational data requirements and time rise proportionately, posing a persistent challenge in CFD. Recent efforts focus on accurately predicting fine-mesh simulations from coarse-mesh data, employing deep learning techniques such as UNets to address this challenge. Existing methods face limitations with unstructured meshes, due to their inability to convolute. Incorporating geometry/mesh information during training brings drawbacks like increased data requirements, and challenges in generalization to unseen geometries. To address these concerns, we propose a novel framework, **PointSAGE** a mesh-independent network that leverages the unordered, mesh-less nature of Pointcloud to learn the complex fluid flow and directly predict fine simulations, completely neglecting mesh information. With an adaptable framework, PointSAGE accurately predicts fine data across diverse point cloud sizes, *regardless of the training dataset's dimension*. Evaluations of various datasets and scenarios demonstrate notable results, showcasing a significant acceleration in computational time for generating fine simulations compared to standard CFD.

## 1 INTRODUCTION

The Navier-Stokes equation offers insight into complex physics, but its non-linearity presents challenges. Computational Fluid Dynamics (CFD) addresses these challenges with various numerical methods. Predicting intricate flows often requires fine resolutions, increasing computational requirements. In Direct Numerical Simulation (DNS), fine simulations demand significant CPU hours and approximately a terabyte (TB) of memory, while coarse simulations require half the time and 1/100th of the memory for the same phenomena (Hawkes et al. (2005)). Hence, coarse grid simulations have become important due to computational efficiency. Yet, the persistent pursuit of understanding complex phenomena drives the ongoing demand for fine-mesh simulations. Recent advancements inspired by super-resolution techniques have introduced deep-learning methodologies for predicting fine-mesh simulations from coarse-mesh counterparts. Utilizing established architectures like MLP (Nair & Goza (2020), U-Nets(Pathak et al. (2020)), and GANs (Xie et al. (2018)), GNNs(Pfaff et al. (2020)) these approaches show promise in overcoming computational challenges to generate fine simulations. Despite recent advancements, current research faces challenges as models follow a traditional super-resolution approach, coarsening fine simulations through down-sampling techniques (Bode et al. (2021); Esmaeilzadeh et al. (2020)). The training bias introduced by this process results in accurate outcomes, but real-life scenarios often involve coarse data not directly down-sampled from the fine counterpart. Researchers are now incorporating actual coarse data, aligning more closely with real-world scenarios (Sarkar et al. (2023)).

However, the current research primarily focuses on regular structured data (Sarkar et al. (2023); Pathak et al. (2020)), limiting the model's adaptability to various data formats, including unstructured or irregular data. When facing unstructured data, these models struggle due to their inability to *undergo convolution*. Integrating mesh information during training adds complexity and data requirements, prolonging training time. Model training on specific geometries restricts predictive capabilities, especially for unseen geometries under same physical phenomena, leading to generalization issues. Accurately obtaining mesh information in real-world scenarios is challenging, com-

promising the model's robustness for precise predictions in fine-mesh simulations. A framework that is independent of geometry/mesh information is crucial. Point clouds provide a comprehensive representation of 3D space by capturing the spatial coordinates of individual points within the domain. In recent times, they have gained prominence due to their ability to handle unstructured data, thanks to their unordered nature (Qi et al. (2017)). This flexibility allows point clouds to capture intricate details and irregularities, making them well-suited for modeling diverse and complex scenarios.

To overcome these challenges, we propose **PointSAGE**, a *mesh-independent* framework that utilizes point clouds derived from entities like cell centers and nodes. This approach provides flexibility for managing irregular and unstructured data due to its unordered nature. Inspired by the "classification network" in *PointNet* (Qi et al. (2017)) and leveraging *SAGEConv* (Hamilton et al. (2018)), PointSAGE captures global and local inter-dependencies within fluid flow features respectively. The model seamlessly predicts fine mesh data, irrespective of training set size/dimensionality, *enabling accurate predictions for any value of* $n$. Extensive testing across diverse datasets and scenarios demonstrates PointSAGE's remarkable performance, with significant reductions in training time compared to state-of-the-art (SOTA) techniques.

## 2  POINTSAGE SUPER-RESOLUTION ON POINT CLOUD

In this section, we present the architecture for super-resolving coarse simulation point cloud data to match fine simulation point cloud data. The model $f : C \to F$ accurately maps the non-linear relationship between the coarse point cloud $C \in \mathbb{R}^{m \times d}$ and the fine point cloud $F \in \mathbb{R}^{n \times d}$, where $m \ll n$. Importantly, our approach is independent of mesh information, focusing solely on state-variable information. The architecture, shown in Figure 1, comprises an Inverse Distance Weighting (IDW) upsampler, a Global Feature Extractor, and a Local Feature Extractor.

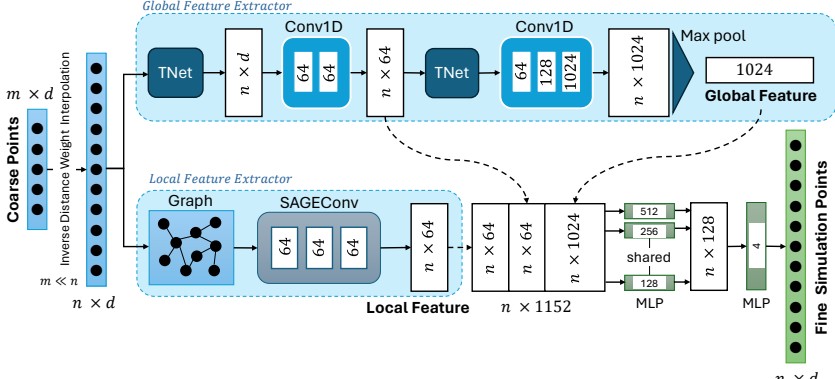

Figure 1: PointSAGE Architecture

**IDW Upsampler:** The coarse point cloud $m$ is up-sampled using Inverse Distance Weighting (IDW) to align its resolution with the fine-point cloud $n$. The interpolated feature $V(n)$ at point $n$ is calculated as:

$$V(n) = (\sum_{i=1}^{n} V_i \cdot e^{-(D_i/D_0)^2}) / \sum_{i=1}^{n} e^{-(D_i/D_0)^2}$$

where $D_i$ is the Euclidean distance between coarse points $m$ and fine points $n$, and $D_0$ is the correlation distance. The interpolated point cloud is then fed into the Global and Local Feature Extractors.

**Global Feature Extractor:** The interpolated point cloud undergoes an input transformation using a TNet followed by two Conv1D layers. Another TNet is applied for feature transformation, and multiple Conv1D layers aggregate features to extract a global feature vector of dimension 1024, inspired by PointNet's classification section.

**Local Feature Extractor:** Simultaneously, the interpolated point cloud undergoes local feature extraction using a graph $G$ construction technique based on a radius near-neighbor approach. Multiple SAGEConv layers are then applied to capture local geometric structures and relationships, resulting in a local feature vector.

The global and local feature vectors are concatenated and passed through MLPs to map the coarse point cloud $C$ to the fine point cloud $F$. The Mean Squared Error is used as the loss function for training. Importantly, our model remains entirely mesh-independent throughout, ensuring robustness and versatility across diverse mesh representations.

## 3 RESULTS AND DISCUSSION

To showcase our model's effectiveness, we experimented with three diverse datasets. We assessed our model's performance using 2D point cloud data from the forward-facing step simulation and extended it to 3D lid-driven cavity simulation data. Additionally, we compared our model to recent work, specifically PIUNet, which focuses on regular grid fluid flow prediction using 2D methane combustion data. This comparison aims to highlight the advantages of our approach, PointSAGE, utilizing point cloud data in the simulation.

**Forward-Step:** The transient CFD simulation of the forward step reveals insights into complex flow phenomena related to separated and reattached flows in a step-like configuration. We focused on predicting the shock wave generated from supersonic flow at the inlet to a rectangular geometry with a step. Details of the simulation are explained in the appendix A.1. To demonstrate our approach's effectiveness, we showcased the model's performance in two scenarios: varying inlet velocity ($U_\infty \in [2, 5]$ m/s) for a fixed Aspect Ratio (AR) of 3, and varying both inlet velocity ($U_\infty \in [2, 5]$ m/s) and Aspect Ratio ($AR \in [3, 6]$). Table 1 indicates PointSAGE outperforms other mesh-independent approaches in both scenarios, achieving approximately 70% and 20% lower RMSE in all features compared to the second-best results in scenarios 1 and 2, respectively.

Table 1: Performance of PointSAGE fine mesh prediction on 2D forward step dataset

**Scenario 1:** Varying inlet velocity for an Aspect Ratio (AR) of 3 (with a 2.4m block length beyond the step)(3352 pts → 42702 pts)

| | Algorithm | MAE | RMSE | $R^2$ | | MAE | RMSE | $R^2$ | | MAE | RMSE | $R^2$ | | MAE | RMSE | $R^2$ |
|---|---|---|---|---|---|---|---|---|---|---|---|---|---|---|---|---|
| $U_x$ | SAGEConv | 0.9025 | 1.0919 | 0.1695 | $U_y$ | 0.2434 | 0.3847 | 0.5572 | Pressure | 2.4264 | 4.5794 | 0.1626 | Mach No. | 0.5665 | 0.7725 | 0.6726 |
| | PointNET | 0.6428 | 1.017 | 0.3126 | | 0.2687 | 0.4065 | 0.4973 | | 2.0413 | 4.0606 | 0.3416 | | 0.5101 | 0.758 | 0.6784 |
| | **PointSAGE** | **0.1651** | **0.3418** | **0.8733** | | **0.0941** | **0.1668** | **0.8996** | | **0.7323** | **1.6079** | **0.8691** | | **0.1476** | **0.275** | **0.9415** |

**Scenario 2:** Training on AR 3 and 4, validating on AR 5, testing on AR 6 (6082 pts → 91302 pts), with varying inlet velocity and aspect ratio.

| | Algorithm | MAE | RMSE | $R^2$ | | MAE | RMSE | $R^2$ | | MAE | RMSE | $R^2$ | | MAE | RMSE | $R^2$ |
|---|---|---|---|---|---|---|---|---|---|---|---|---|---|---|---|---|
| $U_x$ | SAGEConv | 0.6254 | 0.8411 | -0.9743 | $U_y$ | 0.2324 | 0.3138 | -0.4679 | Pressure | 1.6785 | 2.6554 | -0.3859 | Mach No. | 0.667 | 0.8497 | -0.8944 |
| | PointNET | 0.5009 | 0.7747 | -0.2627 | | 0.2052 | 0.2985 | -0.1746 | | 1.3818 | 2.2660 | 0.0150 | | 0.5147 | 0.7196 | -0.0550 |
| | **PointSAGE** | **0.2810** | **0.5008** | **0.3125** | | **0.2007** | **0.2859** | **-0.0962** | | **1.1817** | **2.0587** | **0.0380** | | **0.2854** | **0.4610** | **0.4408** |

**Lid-Driven Cavity:** Hanna et al. (2017) This case study on the lid-driven cavity showcases our model's predictive capabilities. It highlights two main aspects: the model's ability to predict turbulence in the flow, particularly the bottom-right vortex in the cavity Figure 11(available in A.2), and its effectiveness in handling unseen geometries or conditions after training on various scenarios (Table 2). Our model achieves comparable accuracy with existing benchmark techniques in scenarios such as Re interpolation or Re&GS extrapolation, requiring significantly less training time—specifically, **five times faster**. Additionally, we've utilized methods like SAGEConv and PointNet to demonstrate their effectiveness in learning and predicting fine-mesh simulations with notable accuracy. While these methods perform well in simpler datasets, our model excels in managing more complex data scenarios. Additional information about the other scenarios and results are discussed in the Appendix A.2.

**Methane Combustion:** This study compares two methodologies, PointSAGE and recent work as mentioned above, PIUNet (Sarkar et al. (2023)), focusing on methane combustion. Unlike PIUNet, which relies on a regular mesh grid, PointSAGE offers a mesh-independent approach for super-resolution on any mesh or geometry. Table 3 shows that while PointSAGE achieves comparable results in adiabatic temperature ($T_{adia}$), it outperforms or ranks second best in other features, including $U_x$, $U_y$, and mass fractions of $CH_4$, $O_2$, and $CO_2$. These results highlight PointSAGE's capability to predict finer mesh simulation outcomes with comparable accuracy.

From Figure 2, it is clear that PointSAGE significantly reduces computation time compared to traditional CFD simulations. Simulations with PointSAGE achieve impressive speedups, with a *30X, 72X, and 92X* improvement in simulation time for the 2D Forward Step, 3D Lid-driven Cavity, and 2D Methane Combustion cases, respectively. These speedups are crucial for real-world applications

Table 2: Performance of PointSAGE fine mesh prediction on 3D Lid Driven cavity dataset (where Time is Training time)

| | Algorithms | MSE ($1e^{-4}$) | $R^2$ | Time (sec) | | MSE ($1e^{-4}$) | $R^2$ | Time (sec) | | MSE ($1e^{-4}$) | $R^2$ | Time (sec) | | MSE ($1e^{-4}$) | $R^2$ | Time (sec) |
|---|---|---|---|---|---|---|---|---|---|---|---|---|---|---|---|---|
| | CG-CFD | 1 | 0.915 | 660 | | 1 | - | 660 | | 1 | - | 660 | | 1 | - | 780 |
| | UNet | 1.3 | 0.971 | 600 | | 1.4 | 0.944 | 660 | | - | - | - | | - | - | - |
| | SAGEConv | 2.67 | 0.965 | 80 | | 3.6 | 0.933 | 83 | | 3 | 0.959 | 93 | | 1.7 | 0.978 | 41 |
| | PointNET | 2.7 | 0.967 | 26 | | 3.3 | 0.937 | 29 | | 3.1 | 0.961 | 28 | | 1.7 | 0.98 | 13 |
| | **PointSAGE** | 2.6 | 0.959 | 120 | | 3.5 | 0.924 | 120 | | 3 | 0.952 | 156 | | 2.3 | 0.968 | 50 |

(Row labels, left to right sections: **Re Interpolation** / **Re Extrapolation** / **Re & GS Interpolation** / **Re & GS Extrapolation**; Features: $U_x$ for CG-CFD, $U_x, U_y, U_z$ for the learned models.)

Table 3: PointSAGE fine mesh prediction (1000 pts → 50000 pts) on 2D Methane Combustion

| | | Algorithm | MAE | RMSE | $R^2$ | | MAE | RMSE | $R^2$ | | MAE | RMSE | $R^2$ |
|---|---|---|---|---|---|---|---|---|---|---|---|---|---|
| **Fluid properties** | Temperature | UNet | 13.224 | 30.718 | 0.9963 | | 0.0177 | 0.0296 | 0.9839 | | 0.0152 | 0.0324 | 0.9830 |
| | | PIUNet | **10.385** | **20.954** | **0.9984** | | 0.0164 | 0.0286 | **0.9862** | | 0.0158 | 0.0324 | 0.9835 |
| | | SAGEConv | 18.209 | 30.075 | 0.9959 | $U_x$ | 0.0079 | 0.0107 | 0.9924 | $U_y$ | 0.0109 | 0.0153 | **0.9940** |
| | | PointNET | 27.543 | 46.745 | 0.9847 | | 0.0075 | 0.0105 | 0.9836 | | 0.0154 | 0.0222 | 0.9735 |
| | | **PointSAGE** | 17.296 | 28.701 | 0.9947 | | **0.0071** | **0.0105** | 0.9855 | | **0.0104** | **0.0151** | 0.9930 |
| **Mass Fraction** | $CH_4$ | UNet | 0.0140 | 0.0140 | 0.9938 | | 0.0059 | 0.0116 | 0.9870 | | 0.0058 | 0.0094 | 0.9564 |
| | | PIUNet | 0.0138 | 0.0138 | 0.9954 | | 0.0030 | 0.0106 | 0.9888 | | 0.0018 | 0.0051 | **0.9844** |
| | | SAGEConv | **0.0089** | **0.0128** | **0.9987** | $O_2$ | 0.0024 | 0.0041 | 0.9974 | $CO_2$ | 0.0013 | 0.0024 | 0.9639 |
| | | PointNET | 0.0125 | 0.0191 | 0.9932 | | 0.0028 | 0.0053 | 0.9954 | | 0.0016 | 0.0033 | -386.93 |
| | | **PointSAGE** | 0.0100 | 0.0140 | 0.9971 | | **0.0024** | **0.0038** | **0.9975** | | **0.0012** | **0.0022** | 0.9344 |

where computational efficiency is paramount. Furthermore, PointSAGE demonstrates accurate predictions even in unseen scenarios, positioning it as a powerful tool for accelerating CFD simulations while maintaining high prediction accuracy across various domains and scenarios.

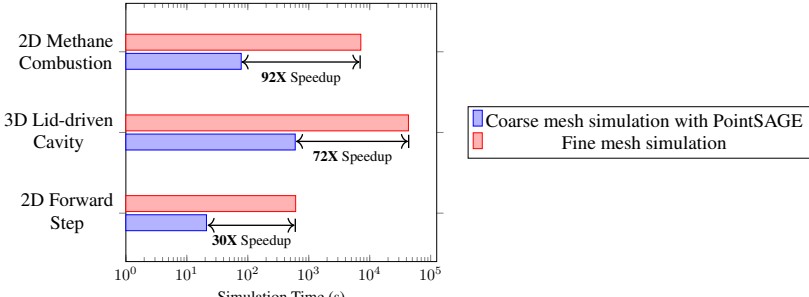

Figure 2: Speedup Comparison of PointSAGE in CFD Simulations: *Blue bars* show the time for coarse mesh simulation with PointSAGE inference for fine mesh prediction, and *red bars* indicate simulation time for fine mesh using OpenFOAM.

## 4 CONCLUSION

We present PointSAGE, a point cloud-based superresolution model capable of predicting fine-mesh data from coarse-mesh input without prior knowledge of mesh characteristics. Leveraging point cloud data, PointSAGE demonstrates robust performance across diverse datasets and unseen geometries, showcasing generalizability and adaptability to different shapes and sizes. In a forward-facing step simulation, our model accurately captures shock formations, achieving substantial improvements in RMSE and MAE compared to existing techniques. Similarly, in Lid-driven cavity simulations, PointSAGE exhibits superior predictive capability in turbulent scenarios within a 3D computational domain, showcasing significant reductions in training time and improved MSE. The key innovation lies in our mesh-independent approach, eliminating the need for detailed mesh information and setting the stage for future advancements in fluid flow simulations.

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

# A    APPENDIX

## A.1    CASE STUDY 1: FORWARD FACING STEP SIMULATION

In this case study we investigate transient simulation of supersonic flow over a forward-facing step using sonicFoam solver in OpenFOAM. The sonicFoam solver is designed to solve compressible trans-sonic/supersonic laminar gas flow. The problem description involves a flow of Mach 3 at an inlet to a rectangular geometry with a step near the inlet region that generates shock waves and propagates downstream and get reflected from the walls and creates reflected shocks in the remaining length after the forward step till the time it reaches its steady state. This case study we have selected from the OpenFOAM tutorial this link.

### A.1.1    PROBLEM DESCRIPTION

**Solution domain**
The 2D computational domain features a step with a height of 20% located at a distance of 0.6m from the inlet as shown in the Figure 3. The experiment is conducted in a gas medium with a speed of sound given by $\sqrt{\gamma RT} = 1$ m/s. Thus, at the inlet, the flow is supersonic with a Mach number of 3 ($U_\infty = 3$ m/s), along with a pressure of 1 Pa and a temperature of 1 K. The aspect ratio of the

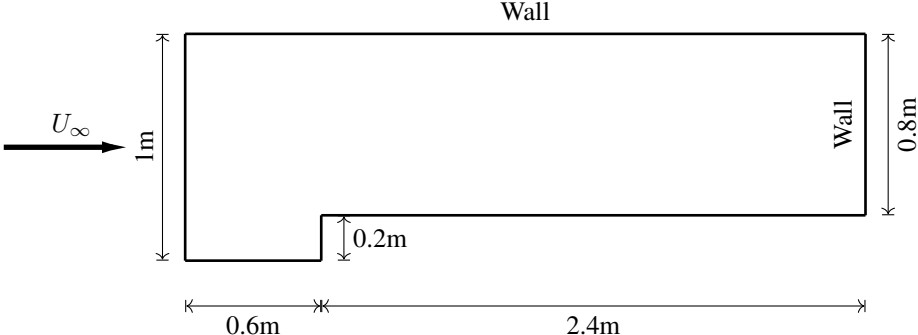

Figure 3: Computational Domain of a 2D Forward Facing Step Simulation

defined geometry in this case study is expressed as the ratio of the length of the rectangular domain (3m) to its height (1m), as shown below:

$$\text{Aspect ratio} = \frac{\text{Length}}{\text{Height}} = \frac{3\,\text{m}}{1\,\text{m}}$$

**Governing equations**

Mass continuity:

$$\frac{\partial \rho}{\partial t} + \nabla \cdot (\rho \mathbf{U}) = 0 \tag{1}$$

Ideal gas:

$$p = \rho RT \tag{2}$$

Momentum equation for Newtonian fluid:

$$\frac{\partial(\rho \mathbf{U})}{\partial t} + \nabla \cdot (\rho \mathbf{U} \mathbf{U}) - \nabla \cdot \mu \nabla \mathbf{U} = -\nabla p \tag{3}$$

The energy equation for fluid (ignoring some viscous terms):

$$\frac{\partial(\rho e)}{\partial t} + \nabla \cdot (\rho \mathbf{U} e) - \nabla \cdot \left( \frac{k}{C_v} \nabla e \right) = p \nabla \cdot \mathbf{U} \tag{4}$$

**Initial Conditions:**

$$U = 0 \, \text{m/s}, \quad p = 1 \, \text{Pa}, \quad T = 1 \, \text{K}$$

**Boundary Conditions:**

- **Inlet (left):**

  FixedValue for velocity: $U = 3 \, \text{m/s} \, (\text{Mach 3})$
  Pressure: $p = 1 \, \text{Pa}$
  Temperature: $T = 1 \, \text{K}$

- **Outlet (right):**

  ZeroGradient on $U, p,$ and $T$

- **No-slip adiabatic wall (bottom)**
- **Symmetry plane (top)**

**Transport Properties:**

$Laminar$               Dynamic viscosity of air: $\mu = 18.1 \, \mu\text{Pas}$

**Thermodynamic Properties:**

Specific heat at constant volume: $C_v = 1.78571 \, \text{J/kgK}$
Gas constant: $R = 0.714286 \, \text{J/kgK}$
Conductivity: $k = 32.3 \, \mu\text{W/mK}$

### A.1.2 MESH DESCRIPTION

The mesh is generated using the *blockMesh* utility, dividing the domain into uniform rectangular cells. For the fine mesh, the cells have dimensions of 0.03 m in the $x$-direction and 0.025 m in the $y$-direction, resulting in 42702 points for the point cloud. Conversely, the coarse mesh divides the domain into cells with dimensions of 0.12 m in the $x$-direction and 0.1 m in the $y$-direction, yielding 3352 points for the coarse point cloud used in our neural network.

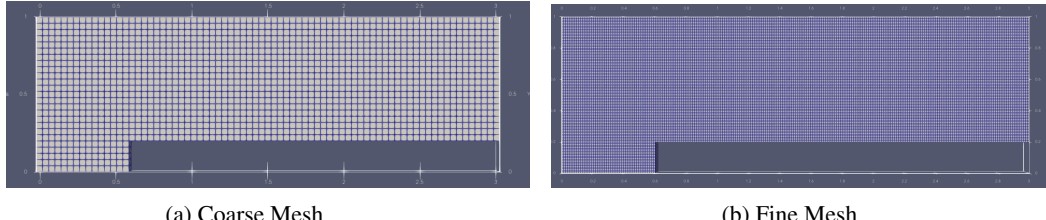

(a) Coarse Mesh                           (b) Fine Mesh

Figure 4: Mesh Description: Aspect Ratio 3

### A.1.3 SCENARIO 1

In this scenario, we conducted experiments using a dataset where the inlet velocity was varied in the range of 2 m/s to 5 m/s, with intervals of 0.25 m/s, while maintaining a constant aspect ratio of 3, as mentioned earlier. The simulations were performed on an Intel(R) Core(TM) i7-8700 CPU @ 3.20GHz. For deep learning experiments, we partitioned the dataset into 80%/10%/10% for training, validation, and testing respectively, and executed the entire experiment on a Tesla P100 GP with 16GB VRAM.

**Results**

The simulations conducted were of a transient nature, and PointSAGE demonstrated commendable accuracy in predicting features such as pressure and velocity at different time intervals, as illustrated in Figure 6 and Figure 7 respectively. Both figures reveal that our PointSAGE model effectively captures the propagation of shocks and their reflection within the rectangular domain following the step location. The training and validation for the PointSAGE training can be observe in this Figure 5.

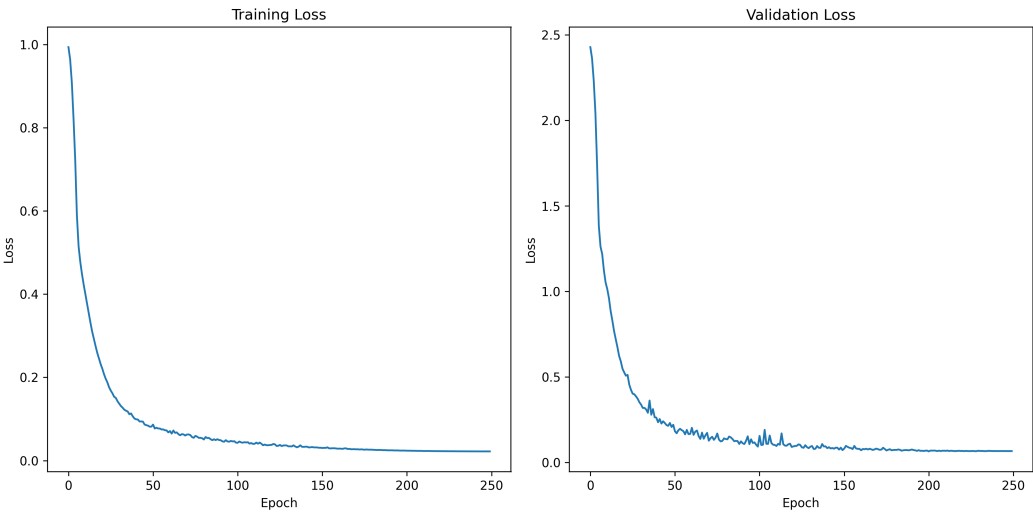

Figure 5: PointSAGE Training and Validation Loss

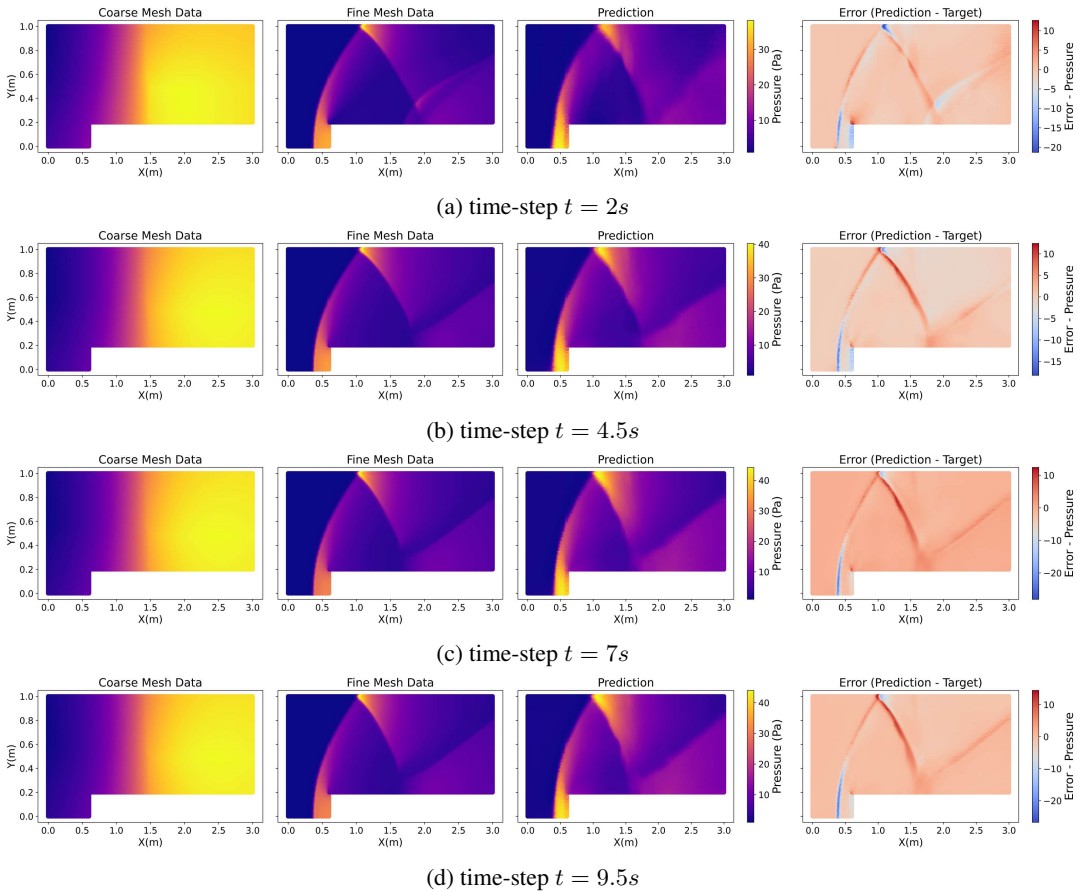

(a) time-step $t = 2s$

(b) time-step $t = 4.5s$

(c) time-step $t = 7s$

(d) time-step $t = 9.5s$

Figure 6: Pressure distribution from PointSAGE-predicted fine mesh simulation at various time steps, corresponding to an inlet velocity $U_\infty$ of 4.875 m/s.

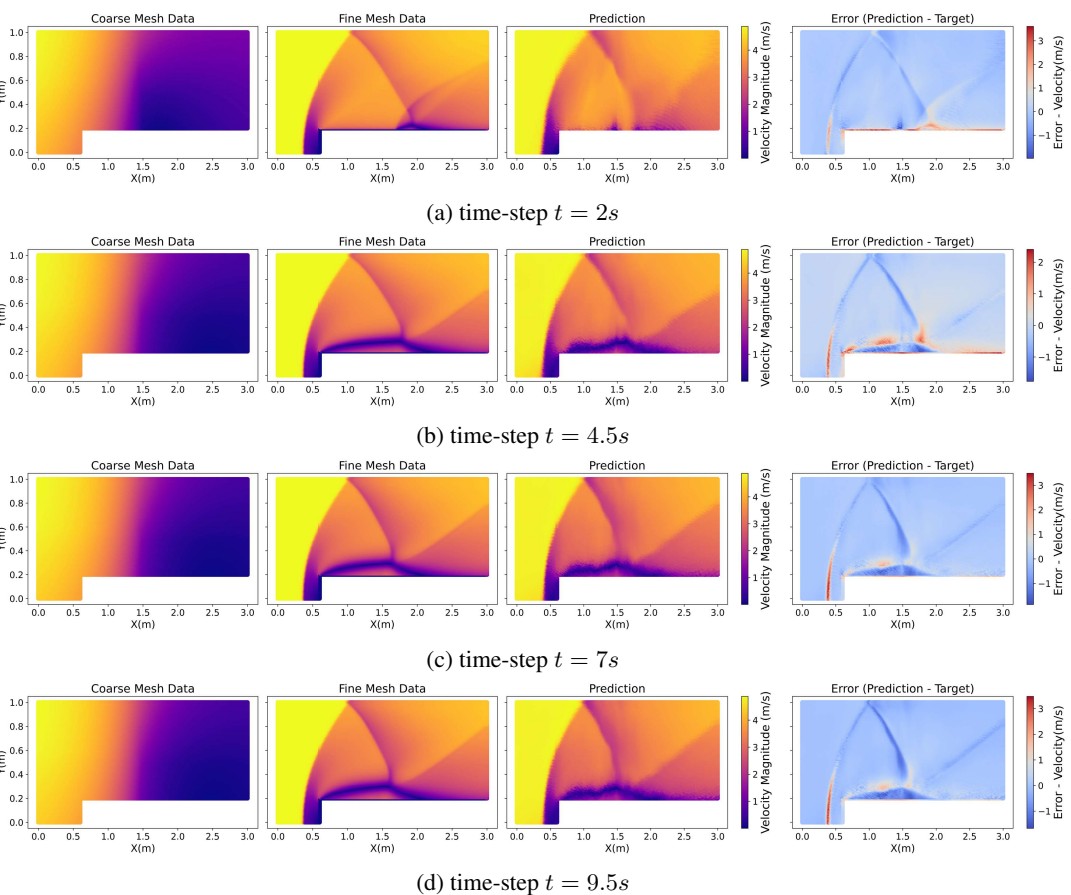

Figure 7: Velocity distribution from PointSAGE-predicted fine mesh simulation at various time steps, corresponding to an inlet velocity $U_\infty$ of 4.875 m/s.

### A.1.4 SCENARIO 2:

In this particular scenario, we conducted experiments utilizing a dataset wherein the inlet velocity was systematically varied within the range of 2 m/s to 5 m/s, with intervals of 0.5 m/s. Additionally, we varied the aspect ratio within the range of 3 to 6. The variation in aspect ratio involves an increase in the length after the step location at 0.6 m. For instance, in the case of an aspect ratio of 3, the length of the section after the step is 2.4 m (total length = 0.6 + 2.4 = 3m). On the other hand, for an aspect ratio of 4, the length of the section after the step is 3.4 m (total length = 0.6 + 3.4 = 4m). In the context of deep learning experiments, we partitioned the dataset for training using aspect ratios 3 and 4, for validation with aspect ratio 5, and for testing with aspect ratio 6.

**Results**

The objective of this scenario is to evaluate the model's proficiency in effectively understanding and adapting to physical phenomena, specifically shock formation and reflection, within a given aspect ratio. Furthermore, the model is challenged to extend its predictions to another aspect ratio, adding complexity as an increase in the length after the step leads to intensified shock reflection and sudden alterations in flow behavior downstream. As illustrated in Figure 8, PointSAGE demonstrates satisfactory predictions for essential features such as pressure and velocity. This success underscores the model's ability to effectively capture and forecast the dynamic behaviors of shocks under varying aspect ratios, emphasizing its efficacy in handling complex flow phenomena.

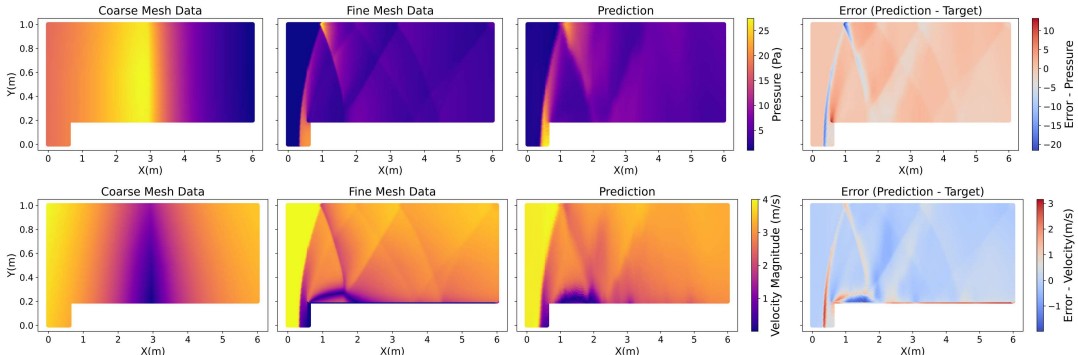

Figure 8: Scenario 2: Training on AR 3 and 4,and pressure and velocity prediction at t= 3.5s for inlet velocity 4 m/s for an AR 6 dataset.

## A.2 CASE STUDY 2: LID-DRIVEN CAVITY

In this case study, we explore the lid-driven cavity, a relatively straightforward scenario compared to others, well known as a benchmark problem in computational fluid dynamics (CFD). The problem entails modeling the fluid flow within a cubic cavity, with a width of 1m and a lid velocity ($U_{lid}$) of 1m/s, resulting in intricate fluid phenomena, notably the formation of counter-rotating vortices at the cavity's bottom. The flow characteristics vary depending on factors such as the Reynolds number and aspect ratio. To simulate this, we employed a transient solver known as *pisoFoam*, implementing the PISO algorithm. For the present work, it is focused on the quasi-steady state flow within the cavity.

### A.2.1 PROBLEM DESCRIPTION

As explained above, the computational domain comprises a 3D cube cavity with a width of 1m, as illustrated in Figure 9. The cavity's height adjusts proportionally based on the aspect ratio. In this study, varying Reynolds numbers are achieved by altering the kinematic viscosity while maintaining a constant $U_{lid}$ value. Specifically, Reynolds numbers of 6000, 8000, 10000, and 12000 are considered, with grid sizes of 1/20, 1/30, and 1/40 employed for simulating coarse mesh data. For fine mesh data, a grid size of 1/120 is utilized. To enhance turbulence capture within the cavity, wall refinement is implemented in the fine mesh data's wall region, as depicted in the Figure 10, whereas such refinement is omitted for coarse data.

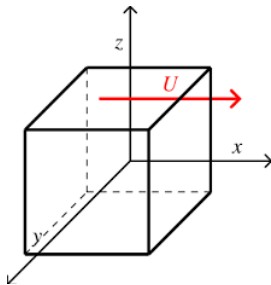

Figure 9: Lid Driven Cavity

Following the generation of fine mesh data, it is overlaid onto the coarse mesh, resulting in identical mesh sizes denoted as $(r, d)$, where $r$ signifies the number of points, and $d$ represents the number of features. In this specific instance, the model's up-sampling aspect is bypassed since the input and output sizes match. In alignment with the source paper's methodology, which evaluated the model's predictability across diverse scenarios, including Reynolds number extrapolation, our study adheres to a similar approach. The model is trained on a subset of Reynolds numbers (e.g., 6000, 80000, 10000) and tested on entirely different ones (e.g., 12000), replicating six such scenarios. Herein, we concentrate on the initial four scenarios to showcase our model's versatility.

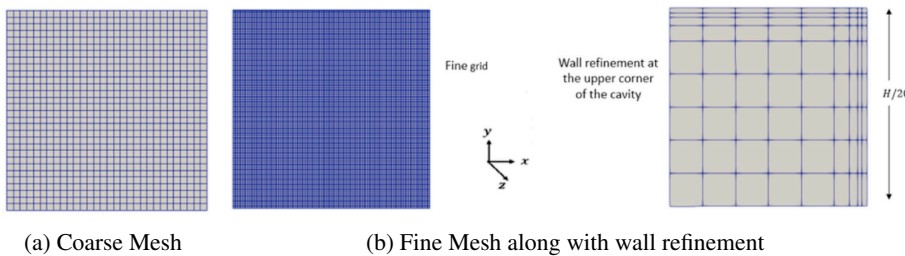

(a) Coarse Mesh        (b) Fine Mesh along with wall refinement

Figure 10: The above figures depict the computational grids utilized in this work

**Results:** To demonstrate the model's effectiveness and predictive ability, we adhere to the protocol outlined in the source paper (Hanna et al. (2017)). In the six scenarios, we will execute the first four scenarios. In these scenarios, the model's hyper-parameters consist of **r**, representing the radius of the graph, and **lr**, indicating the learning rate. Following several iterations, it was observed that the most effective values for "k" and "lr" are 0.005 and 0.001, respectively. Furthermore, each model undergoes training for 300 epochs. In **Scenario - 1**, Reynolds number interpolation is depicted, where training and validation are conducted for flow at Reynolds numbers of **6000, 8000, and 12000**, and the model is tested at an unseen Reynolds number, **10000**. The grid size (1/30) and aspect ratio (1) are kept constant. Figure 11 demonstrates the model's ability to accurately capture the turbulence of the fluid flow, evident from the bottom right corner of the "Prediction" image. The model can predict fine mesh data with a MSE of 2.6-e4 in just 120 seconds. Figure 12 showcases the training and validation loss.

**Scenario - 2:** This scenario illustrates Reynolds number extrapolation, where training and validation are conducted for flow at **Re - 6000, 8000, 10000**, and the model is subsequently tested at a different Reynolds number, **12000**. The grid size (1/30) and aspect ratio (1) remain constant. Figure 13 showcases the model's predictive prowess, with velocity contours plotted. The model accurately captures the turbulent nature of the flow, achieving an MSE of 3.5e-4 within just 120 seconds. Figure 14 showcase the training and validation loss.

In this case, since the grid size and the aspect ratio remain constant, the number of points in the input/output point cloud remains constant, i.e., 27,000. However, if we alter either of them, the number of points changes. In the next two scenarios, the grid size is modified, leading to different point cloud dimensions for the training and testing datasets.

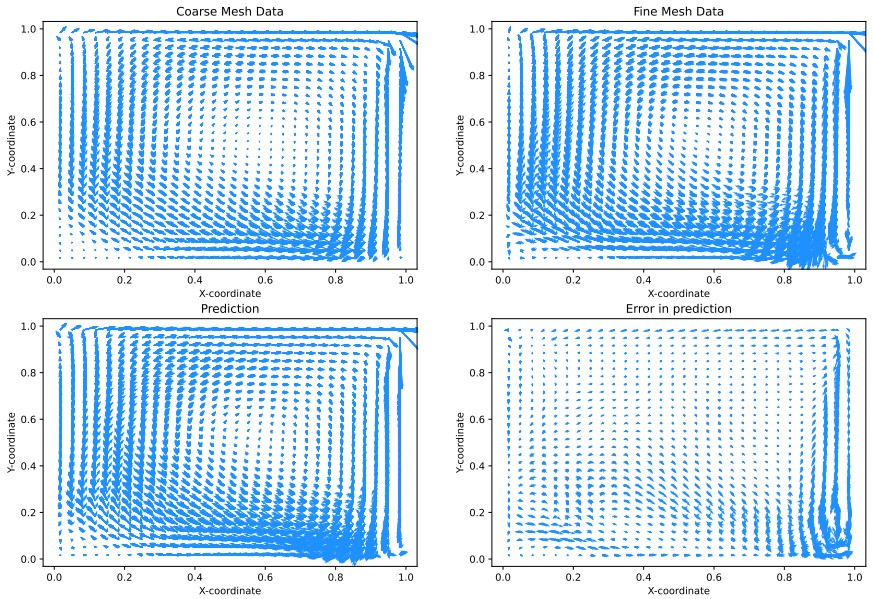

Figure 11: Velocity Contour Plot - Re Interpolation

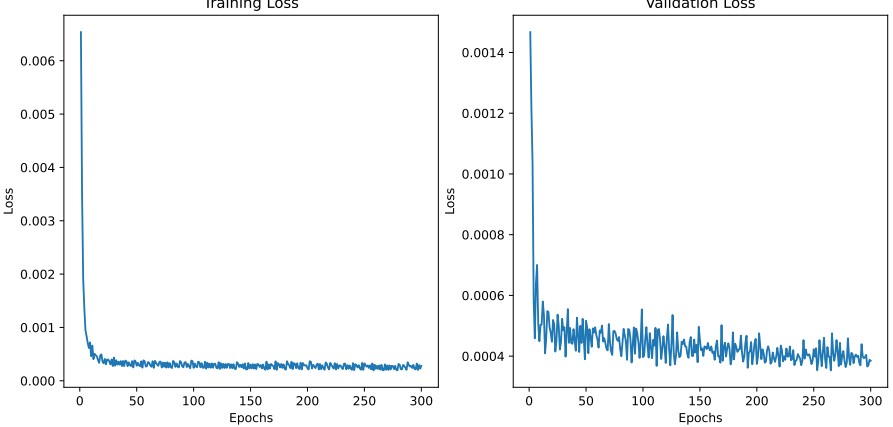

Figure 12: Training and Validation Loss

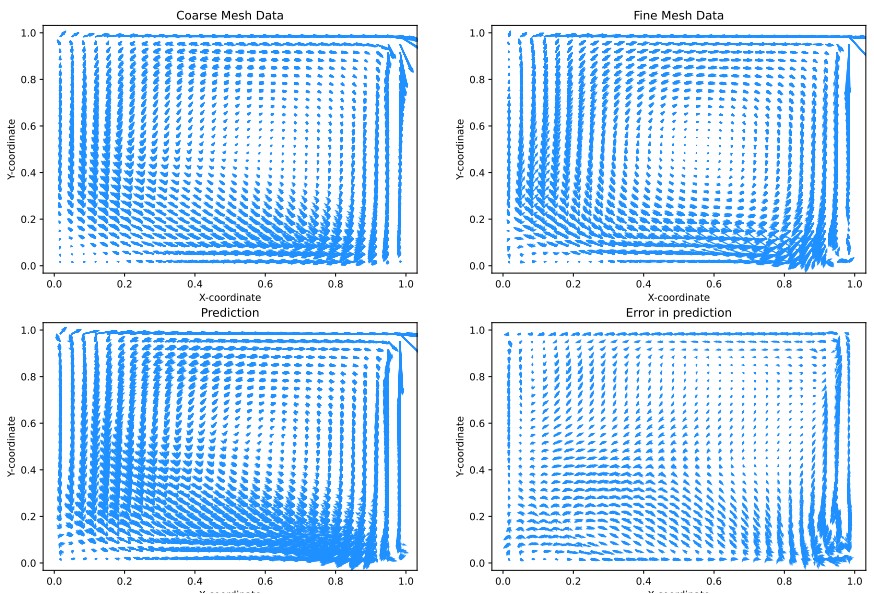

Figure 13: Velocity Contour Plot($U_x$ and $U_y$): Scenario - 2

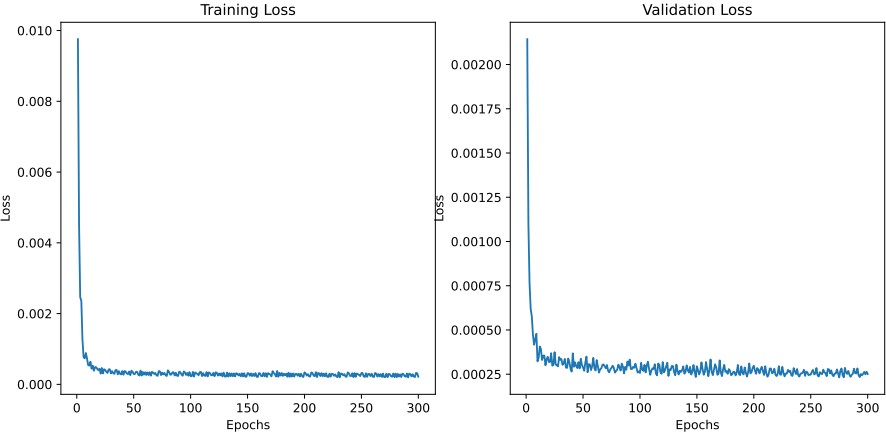

Figure 14: Training and Validation Loss

In **Scenario - 3**, Reynolds number and Grid Size interpolation are employed, where training and validation encompass flow conditions at **Re - 8000, 12000 and Grid size - 1/40, 1/20**. The model is then assessed with a different parameter set, **Re - 10000 and Grid Size - 1/30**. The point cloud dimensions for training, validation, and testing datasets are 64,000, 8,000, and 27,000, respectively, showcasing the model's adaptability to various mesh dimensions. Figure 15 illustrates the velocity contour plot, highlighting the model's adeptness in accurately capturing turbulence, achieving an MSE of 3e-4 within a mere 156 seconds. Figure 16 showcase the training and validation loss.

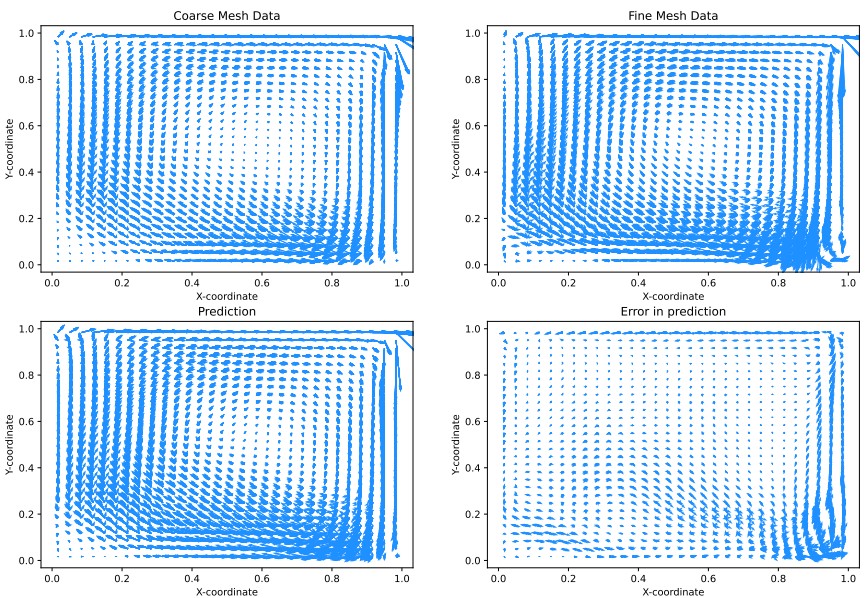

Figure 15: Velocity Contour Plot($U_x$ and $U_y$): Scenario - 3

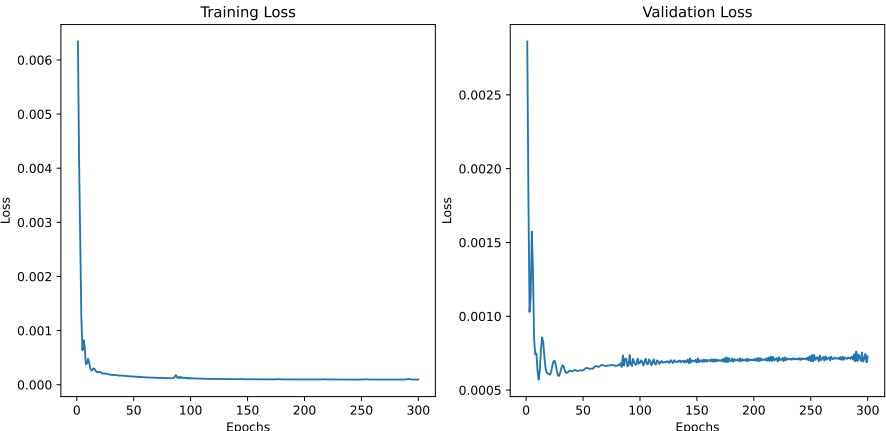

Figure 16: Training and Validation Loss

**Scenario - 4:** In this scenario, both Reynolds number and grid size are varied for training and validation, covering flow conditions at **Re - 8000, 10000 and Grid size - 1/30, 1/20**. The model is then tested with a different parameter set, **Re - 12000 and Grid Size - 1/40**. The point cloud dimensions for training, validation, and testing datasets are 64,000, 8,000, and 27,000, respectively. This scenario presents a greater challenge compared to the previous one. Figure 18 demonstrates

the model's efficacy in accurately capturing turbulence with an MSE of 2e-4 within just 50 seconds. Figure 17 showcase the training and validation loss.

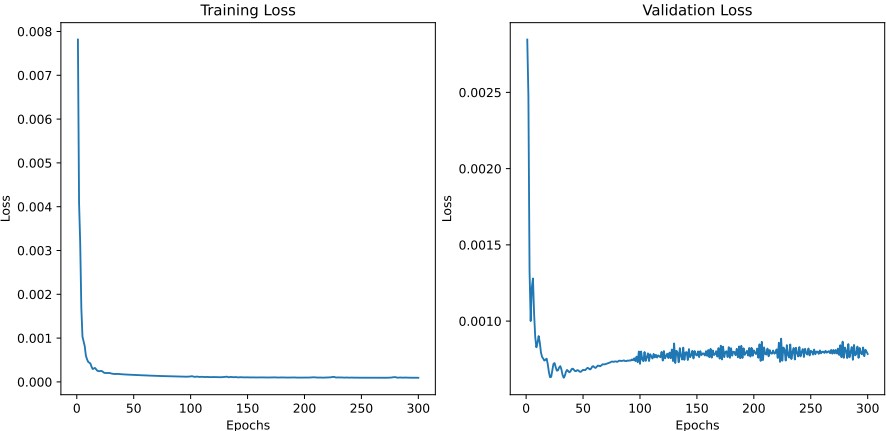

Figure 17: Training and Validation Loss

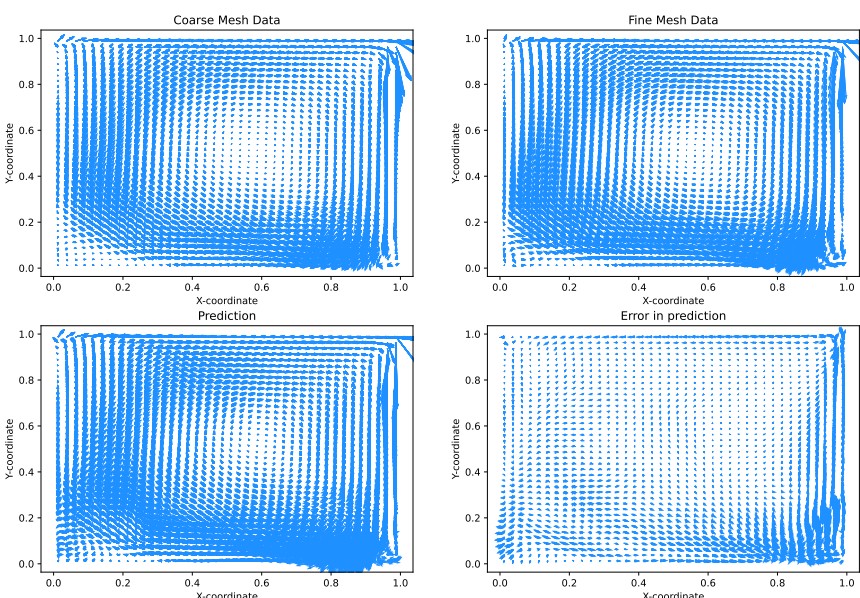

Figure 18: Velocity Contour Plot($U_x$ and $U_y$): Scenario - 4

