# OpenReview forum: "PointSAGE: Mesh-independent superresolution approach to fluid flow predictions"
_ICLR.cc/2024/Workshop/AI4DiffEqtnsInSci — AI4DiffEqtnsInSci @ ICLR 2024 Poster_

### Official Review · Reviewer_t58R · 2024-02-20
**A good paper and interesting for community, however, there are some serious questions that needs to be answered before acceptance.**

**Rating:** 7
**Confidence:** 3

**Review:**

The paper introduces an approach for more efficient prediction of fluid flow, by downscaling the mesh sizes in CFD simulations. Both the approach and results are interesting and should be useful for the community. I believe the work is of a good quality to be accepted for the workshop. However, there are some major (as well as some minor) comments I would like to address before final acceptance:


•	The authors claimed the model is mesh-independent, however, it is not very clear why it is. Maybe a few sentences will help.
•	Regarding the previous comments, as I see in the appendices (e.g., Figs. 4, and 10), none of the cases use unstructured meshes, which is in contradiction to one of the main claimed novelties of the work.
•	The separation of data for training of testing has not been discussed clearly (although I see a discussion about it in the appendix). It would be nice if the authors could mention how they were separated. What was the difference between the properties of the system comparing the training and testing data? that is important since if for instance the geometry or boundary conditions were different between two datasets, the outcomes of the study would be a higher quality.
•	Regarding the previous comment, what are the extrapolation capabilities of the PointSAGE method? Is it reliable to train it once and then use it for many other cases?
•	There are some problems with citations. Maybe authors could use \citep instead of \cite in the cases with parenthetical citations.
•	Table 2 shows that for some cases the PointNet approach is superior considering both accuracy and run-time. What do the authors think about it?

---

### Official Review · Reviewer_ejX4 · 2024-02-28
**PointSAGE: Mesh-independent superresolution approach to fluid flow predictions**

**Rating:** 4
**Confidence:** 4

**Review:**

With the proposed method, the authors are predicting the fine mesh solution from the coarser mesh solution. Once the mapping is learned, the trained networks can be applied with varying field variables (say, velocity) or geometric shapes.

In my opinion, the obtained results are not good, especially for the Euler equations, where it is important to capture shock waves with good accuracy and at exact locations. The author(s) needs to work harder before solving such problems.

---

### Meta-Review · Area_Chair_sSwV · 2024-03-02

**Recommendation:** Accept (Poster)

**Metareview:**

The authors present in their paper PointSAGE, a point cloud-based superresolution model capable of predicting fine-mesh
data from coarse-mesh input without prior knowledge of mesh characteristics. It is interesting work that has been tested on numerous test-cases but the work is still preliminary in nature, given that the three cases are laminar flows and it's not clear how well this would scale to more complex unstructured 3D flows that are turbulent. Nevertheless the work is interesting and it should be accepted for a poster session.

---

### Decision · Program_Chairs · 2024-03-02

Accept (Poster)